# CRT-Fusion: Camera, Radar, Temporal Fusion Using Motion Information for 3D Object Detection

**Jisong Kim**[1,*]   **Minjae Seong**[1,*]   **Jun Won Choi**[2,†]
[1]Hanyang University    [2]Seoul National University
{jskim, mjseong}@spa.hanyang.ac.kr
{junwchoi}@snu.ac.kr

## Abstract

Accurate and robust 3D object detection is a critical component in autonomous vehicles and robotics. While recent radar-camera fusion methods have made significant progress by fusing information in the bird's-eye view (BEV) representation, they often struggle to effectively capture the motion of dynamic objects, leading to limited performance in real-world scenarios. In this paper, we introduce CRT-Fusion, a novel framework that integrates temporal information into radar-camera fusion to address this challenge. Our approach comprises three key modules: Multi-View Fusion (MVF), Motion Feature Estimator (MFE), and Motion Guided Temporal Fusion (MGTF). The MVF module fuses radar and image features within both the camera view and bird's-eye view, thereby generating a more precise unified BEV representation. The MFE module conducts two simultaneous tasks: estimation of pixel-wise velocity information and BEV segmentation. Based on the velocity and the occupancy score map obtained from the MFE module, the MGTF module aligns and fuses feature maps across multiple timestamps in a recurrent manner. By considering the motion of dynamic objects, CRT-Fusion can produce robust BEV feature maps, thereby improving detection accuracy and robustness. Extensive evaluations on the challenging nuScenes dataset demonstrate that CRT-Fusion achieves state-of-the-art performance for radar-camera-based 3D object detection. Our approach outperforms the previous best method in terms of NDS by $+\mathbf{1.7}\%$, while also surpassing the leading approach in mAP by $+\mathbf{1.4}\%$. These significant improvements in both metrics showcase the effectiveness of our proposed fusion strategy in enhancing the reliability and accuracy of 3D object detection.

## 1   Introduction

3D object detection plays a crucial role in autonomous vehicles and robotics, leveraging sensors such as lidar, cameras, and radar to localize and classify objects in the environment. Extensive research has been conducted to explore various strategies for improving detection accuracy and robustness. One prominent approach is the integration of data across multiple timestamps, which aims to mitigate the inherent limitations associated with relying solely on instantaneous data. By incorporating historical information, this approach provides a more robust perception of the environment, addressing the challenges of incomplete data caused by occlusions, sensor failures, and other factors.

Numerous studies have investigated the utilization of temporal information to enhance the performance of LiDAR-based and camera-based 3D object detection methods. Recent works have also explored the incorporation of temporal cues in radar-camera fusion methods [1, 2]. These methods generated bird's-eye view (BEV) feature maps for each frame by fusing radar and camera data into a

---

*Equal contribution.
†Corresponding author.

38th Conference on Neural Information Processing Systems (NeurIPS 2024).

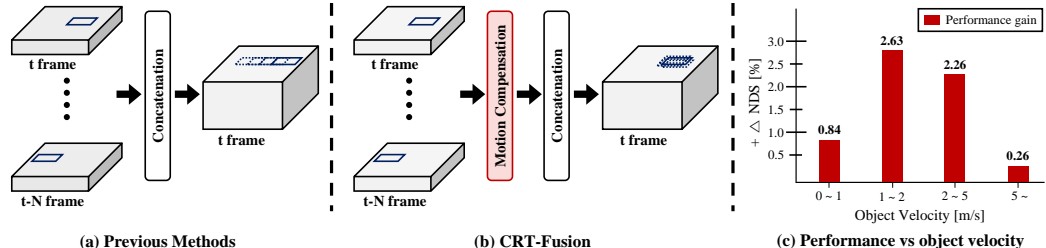

**(a) Previous Methods**      **(b) CRT-Fusion**      **(c) Performance vs object velocity**

Figure 1: **Comparison of temporal fusion methods:** (a) Previous methods concatenate BEV feature maps without considering object motion. (b) CRT-Fusion estimates and compensates for object motion before concatenation. (c) Performance gain of CRT-Fusion over the direct concatenation method, showing CRT-Fusion's superior accuracy across different object velocity ranges.

unified BEV representation. The resulting BEV feature maps are then concatenated across frames to create a comprehensive spatio-temporal representation, as illustrated in Figure 1(a). However, these approaches face limitations in effectively capturing object motion, as they merge data from different time intervals without explicitly considering the dynamics of moving objects. Consequently, the performance accuracy for dynamic objects is compromised.

To address these challenges, we propose a motion-aware approach, as illustrated in Figure 1 (b), which goes beyond simple concatenation of BEV feature maps. Our method first estimates the locations of dynamic objects with their corresponding velocity vector for each timestamped BEV feature map. Subsequently, we leverage this predicted information to rectify the motion of dynamic objects in each feature map and fuse them in a temporally consistent manner. Figure 1(c) presents a graph depicting the performance gain achieved by our proposed method over the direct concatenation of temporal BEV feature maps for different object velocity ranges. It is evident that our approach consistently outperforms the existing method across all velocity ranges, with a notable performance improvement for objects moving at medium velocities. This demonstrates the effectiveness of our motion-aware fusion strategy in capturing and compensating for object motion, leading to superior performance in 3D object detection.

In this paper, we introduce CRT-Fusion, a novel approach for integrating temporal information into radar-camera fusion. Our framework comprises three modules: *Multi-View Fusion* (MVF), *Motion Feature Estimator* (MFE), and *Motion Guided Temporal Fusion* (MGTF). The MVF module generates radar-camera fused BEV feature maps for each timestamp. The MVF enhances image features with radar BEV features, achieving more precise depth predictions through Radar-Camera Azimuth Attention (RCA). The enhanced camera BEV features and radar BEV features are integrated through a gating operation. The MFE module predicts velocity information and performs BEV segmentation for each pixel in the fused BEV features to identify object regions and provide values for shifting the feature map spatially. Finally, the MGTF module generates the final feature map by leveraging the fused BEV feature maps, segmentation results, and velocity predictions. The MGTF module begins with the BEV features from the $(t - N)$th time step and aligns them with those from each subsequent time step. These aligned features are then aggregated one-by-one across all $N$ timestamps in a recurrent manner. Consequently, CRT-Fusion achieves state-of-the-art performance on the nuScenes 3D object detection benchmark for radar-camera fusion methods, with improvements of $+1.7\%$ in NDS and $+1.4\%$ in mAP compared to existing state-of-the-art approaches.

In summary, the main contributions of this work are as follows:

- We introduce CRT-Fusion, a novel framework that effectively integrates temporal information into radar-camera fusion for 3D object detection. By considering the motion of dynamic objects, CRT-Fusion significantly improves detection accuracy and robustness in complex real-world scenarios.

- We design a Multi-View Fusion module that enhances depth prediction by leveraging radar features to improve image features before fusing them into a unified BEV representation.

- We introduce an effective temporal fusion strategy through MFE and MGTF modules. MFE estimates pixel-wise velocity information, and MGTF iteratively aligns and fuses feature maps across multiple timestamps using the motion information obtained from MFE.

- CRT-Fusion achieves state-of-the-art performance on the nuScenes dataset for radar-camera-based 3D object detection, surpassing previous best method by $+1.7\%$ in NDS and $+1.4\%$ in mAP.

## 2 Related Works

### 2.1 Camera-Radar 3D Object Detection

Due to the high cost of LiDAR sensors, research on 3D object detection using radar-camera sensor fusion has gained significant traction in recent years. These studies aim to utilize radar sensors as auxiliary sensors to overcome the fundamental limitation of camera-based 3D object detection research, which is the lack of depth information.

CenterFusion [3] adopts a frustum-based approach to establish connections between radar point clouds and image features, enabling the refinement of 3D proposals. CRAFT [1] captures the interactions between radar and camera data within a polar coordinate system using a cross-attention mechanism, effectively integrating information from both modalities. RCM-Fusion [4] combines radar and image features at both feature-level and instance-level to achieve more precise 3D object detection. RADIANT [5] fuses radar and image features within image pixel coordinates to provide more accurate 3D location estimation. CRN [6] employs a 3D object detection method that strikes a balance between speed and performance by leveraging radar information to enhance camera BEV features and fusing multi-modal BEV features. RCBEVDet [2] introduces a novel radar backbone network that utilizes point-based feature extraction techniques for radar and fuses radar and image features using deformable cross-attention. CRKD [7] employs a method to transfer the knowledge possessed by the LiDAR-Camera fusion detector to the Camera-Radar fusion detector using the Cross-modality Knowledge Distillation technique.

### 2.2 Temporal fusion in 3D Object Detection

The approach to utilizing temporal information in 3D object detection varies depending on the type of sensors employed. In LiDAR-based methods, relying solely on single-frame point cloud data introduces challenges such as occlusion and partial views. To mitigate these issues, several studies have integrated temporal information at the feature level [8, 9, 10, 11]. Another approach involves extracting proposals using an object detector and subsequently leveraging temporal information at the object level [12, 13, 14]. A third method focuses on fusing temporal information within the query representation [15].

On the other hand, camera-based approaches have exploited temporal information to overcome the inherent limitations of image data, such as inaccuracies in depth prediction. One common methodology in camera-based perception research is to fuse temporal information with BEV-based methods [16, 17, 18, 19, 20]. Another approach has centered on enhancing the accuracy of depth estimation through temporal-stereo methods [21, 22, 20].

In the context of radar-camera fusion methods, the utilization of temporal information remains less explored, with most studies following the strategies employed in BEV-based camera-only methods. However, our proposed CRT-Fusion deviates from existing research methods by introducing a novel temporal fusion method. Our approach incorporates a temporal BEV fusion mechanism that explicitly considers the movement of objects, thereby enhancing object detection performance.

## 3 CRT-Fusion

In this work, we introduce CRT-Fusion, a novel framework for 3D object detection that efficiently fuses information from radar, camera, and temporal domains. The overall architecture of CRT-Fusion is illustrated in Figure 2. Our approach first extracts the features from radar and camera data separately using their respective backbone networks. Subsequently, we employ the MVF module to generate

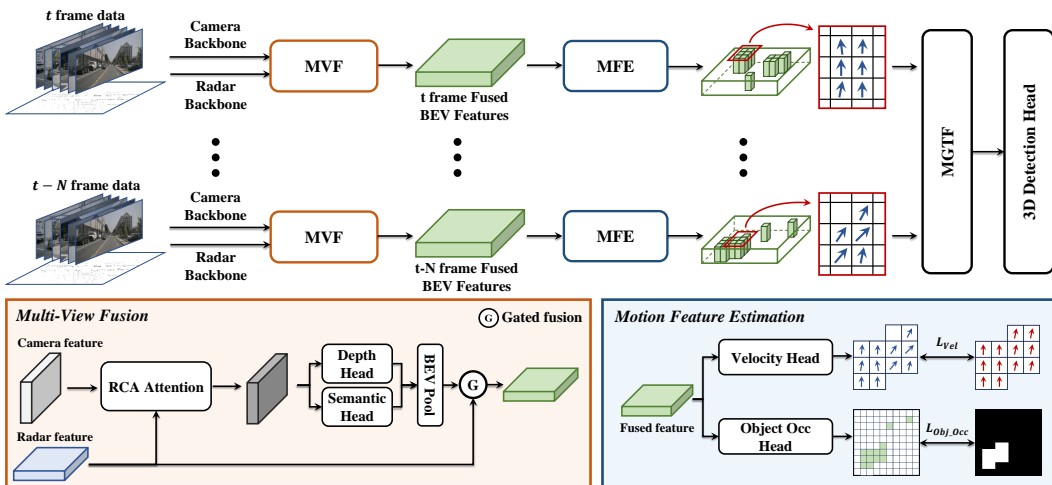

Figure 2: **Overall architecture of CRT-Fusion:** Features are extracted from radar and camera data using backbone networks at each timestamp. The MVF module combines these features to generate fused BEV feature maps. The MFE module predicts the location and velocity of dynamic objects from these maps. The MGTF module then uses the predicted motion information to create the final feature map for the current timestamp, which is fed into the 3D detection head.

a fused BEV feature map for each timestamp by combining the radar and camera features. The sequence of fused feature maps is then utilized by the MFE module to predict the location and velocity information of dynamic objects. The predicted motion information is exploited by the MGTF module to align the BEV feature maps spatially within a time window. Then, the aligned features maps are aggregated to obtain the final feature maps. Finally, 3D object detection is performed using the detection head proposed by CenterPoint [23].

## 3.1 Multi-View Fusion

Recent advancements in BEV-based camera-only approaches have significantly improved performance, leading to an increased focus on radar-camera fusion approaches within BEV representations. These studies [6, 4, 2] primarily aimed to address the inherent limitations of camera-only approaches, particularly the challenge of accurate depth prediction, by leveraging radar data. The existing state-of-the-art model [6] enhanced this process by combining occupancy information from radar point clouds with camera frustum features, effectively incorporating radar positional data. While this method facilitates the direct integration of radar positional information, noise in radar point clouds can adversely affect depth prediction accuracy. To mitigate this issue, we propose a novel fusion strategy that incorporates radar and camera features in both bird's eye view and perspective view. Unlike the existing method [6], our approach enhances camera features using radar information prior to depth prediction, enabling more accurate camera BEV features. Subsequently, we employ a CNN-based gated fusion network to obtain the final fused features.

**Perspective view fusion.** As illustrated in Figure 3 (a), the Radar-Camera Azimuth attention (RCA) module takes the camera perspective view features $F_c \in \mathbb{R}^{N \times C \times H \times W}$ and the radar BEV features $F_r \in \mathbb{R}^{C \times X \times Y}$ as inputs, where $H$ and $W$ represent the height and width of the camera features, and $X$ and $Y$ denote the size of the radar BEV features along the x-axis and y-axis, respectively. For the $i$-th image, the camera feature $F_c^i \in \mathbb{R}^{C \times H \times W}$ is compressed along both height and width dimensions using max pooling and MLP layers, resulting in $W_c^i \in \mathbb{R}^{C \times 1 \times W}$ and $H_c^i \in \mathbb{R}^{C \times H \times 1}$. Let $W_c^i(j)$ be the value of the $W_c^i$ features at the $j$-th position along the width direction. We associate the radar feature element $F_r(x, y)$ at the position $(x, y)$ with $W_c^i(j)$ through Azimuth Grouping. The azimuth angle values corresponding to $W_c^i(j)$ and $F_r(x, y)$ are denoted as $\theta_c^i(j)$ and $\theta_r(x, y)$, respectively. A set of $M$ radar features $\mathcal{R}_j^i$ associated with $W_c^i(j)$ is obtained using:

$$\mathcal{R}_j^i = \left\{ F_r(x, y) \mid \underset{x \in [0, X], y \in [0, Y]}{\arg\min} \left( |\theta_c^i(j) - \theta_r(x, y)|, M \right) \right\} \quad (1)$$

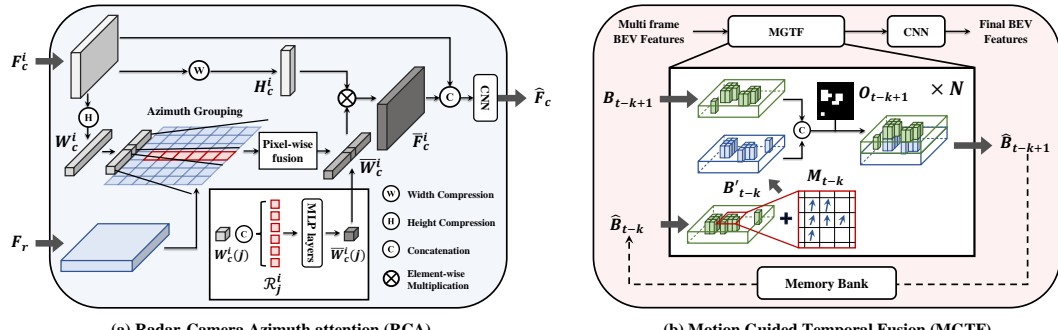

| (a) Radar-Camera Azimuth attention (RCA) | (b) Motion Guided Temporal Fusion (MGTF) |
|---|---|

Figure 3: **Core components of CRT-Fusion:** (a) RCA module enhances image features with radar features for accurate depth prediction. (b) MGTF module compensates for object motion across multiple frames, producing the final BEV feature map for 3D object detection.

where $\arg\min(\mathcal{X}, M)$ returns the indices of the smallest $M$ elements in $\mathcal{X}$, and $\mathcal{R}^i_j$ represents $M$ radar features $F_r(x, y)$ whose azimuth angles are closest to $\theta^i_c(j)$. Pixel-wise fusion module is then applied to $\mathcal{R}^i_j$ to obtain an enhanced feature $\bar{W}^i_c(j)$ as

$$M^i_j(m) = \text{MLP}_2(\text{MLP}_1(\text{concat}(W^i_c(j), \mathcal{R}^i_j(m)))) \tag{2}$$

$$\bar{W}^i_c(j) = \sum_{m=1}^{M} \text{softmax}(\text{MLP}_3(M^i_j(m))) \cdot M^i_j(m) \tag{3}$$

where $\mathcal{R}^i_j(m)$ denotes the $m$-th element of $\mathcal{R}^i_j$. The concatenation of $W^i_c(j)$ and $\mathcal{R}^i_j(m)$ is passed through two MLP layers to obtain an intermediate feature $M^i_j(m)$. These intermediate features are then processed through another MLP layer followed by a softmax function to determine the relevance to $W^i_c(j)$. The weighted sum of these intermediate features yields the enhanced $\bar{W}^i_c(j)$. Finally, we perform an element-wise multiplication of $\bar{W}^i_c \in \mathbb{R}^{C \times 1 \times W}$ and $H^i_c \in \mathbb{R}^{C \times H \times 1}$ to obtain the camera perspective view features $\bar{F}^i_c$. The concatenation of $\bar{F}^i_c$ and $F^i_c$ is then passed through a convolution layer to obtain $\hat{F}^i_c$, which is the perspective view features for the $i$-th image fused with the radar BEV features. These steps are depicted in Figure 3 (a).

**Bird's eye view fusion.** The enhanced camera feature $\hat{F}^i_c$ in the perspective view is employed for depth prediction and camera view semantic segmentation. Inspired by SA-BEV [24], we utilize a 1x1 CNN layer head structure to predict both depth map and segmentation scores in the perspective view. The network outputs $D^i_c \in \mathbb{R}^{(b+1) \times H \times W}$, where $b$ denotes the number of depth bins and the additional dimension corresponds to the foreground score prediction. The predicted segmentation scores are thresholded using a value $\tau_P$ to identify the foreground regions only, which are then projected to the BEV domain. The resulting camera BEV features are fused with the radar BEV features obtained from the radar pipeline using a gated fusion network [25, 26] to yield the final BEV features $B$. The gated fusion network assigns weights to each feature according to their significance, effectively boosting the effect of feature fusion.

### 3.2 Motion Feature Estimation

Temporal fusion methods that consider object motion have been extensively studied in the context of 3D object detection [12, 13, 14]. These methods typically predict object locations and velocities at each timestamp using an object detector, and then aggregate object information from the past timestamps to the current timestamp at the object level. However, this approach has a drawback as the overall model performance heavily depends on the initial detector's performance. Moreover, object-level temporal fusion methods have not utilized motion information of objects effectively. To address these limitations, we propose a simple yet effective solution: by predicting velocity information and object presence for each pixel in the BEV features, the model can produce the aligned BEV features, which can be temporally fused at the feature level in the BEV domain rather than at the object level.

Suppose that we obtain a set of BEV features $\mathcal{B} = \{B_{t-k} | k \in \{0, 1, ..., N\}\}$ from the preceding Multi-View Fusion module, where $B_{t-k}$ represents the BEV feature at timestamp $t - k$. Each feature $B_{t-k}$ is then processed in parallel by two distinct heads: a velocity head and an object occupancy head, both composed of 3x3 and 1x1 convolutions. The velocity head extracts motion information $M_{t-k} \in \mathbb{R}^{2 \times X \times Y}$, containing velocity estimates along the x and y axes for each pixel in the feature map $B_{t-k}$. Simultaneously, the object occupancy head produces an occupancy score map $O_{t-k} \in \mathbb{R}^{1 \times X \times Y}$, where each element indicates whether the corresponding pixel belongs to an object. To facilitate training of these modules, we generate ground truth targets for each timestamp using the following equations. The IoU ratio $r(x, y)$ for a pixel at location $(x, y)$ is defined as:

$$r(x, y) = \frac{|H(x, y) \cap \mathcal{P}(\mathcal{G})|}{|H(x, y)|} \tag{4}$$

where $H(x, y)$ is the physical box in the BEV domain corresponding to one pixel located at $(x, y)$, $\mathcal{G}$ is the set of ground truth 3D object boxes, and $\mathcal{P}(\mathcal{G})$ is the projection of these boxes onto the BEV domain. The IoU ratio calculates the overlap between the physical box and the projected ground truth boxes, helping to determine positive samples. The ground truth values are then given by:

$$M_{t-k}^{GT}(x, y) = \begin{cases} (v_x^{gt}, v_y^{gt}) & \text{if } r(x, y) \geq \tau_{iou} \\ (0, 0) & \text{otherwise} \end{cases} \tag{5}$$

$$O_{t-k}^{GT}(x, y) = \begin{cases} 1 & \text{if } r(x, y) \geq \tau_{iou} \\ 0 & \text{otherwise} \end{cases}, \tag{6}$$

where $M_{t-k}^{GT}(x, y)$ and $O_{t-k}^{GT}(x, y)$ are the ground truth velocity and occupancy scores for pixel $(x, y)$ in the BEV feature $B_{t-k}$, respectively. The velocity vector of the corresponding ground truth object is denoted by $(v_x^{gt}, v_y^{gt})$, and $\tau_{iou}$ is a predefined threshold set to 0.5. Pixels with an IoU ratio exceeding $\tau_{iou}$ are classified as positive and assigned the GT velocity and GT occupancy state for supervision.

## 3.3 Motion-Guided Temporal Fusion

Figure 3 (b) presents the Motion-Guided Temporal Fusion (MGTF) module, which integrates BEV features to construct a dynamic representation of object motion across multiple timestamps. Our model utilizes a memory bank structure, where previously computed BEV features generated through the MGTF process are stored in the buffer, effectively minimizing redundant computations. This memory-efficient design significantly reduces the computational overhead during temporal fusion.

At each timestep $t-k$, the BEV feature map $\hat{B}_{t-k}$ is retrieved from the memory bank, representing the previous result processed through the MGTF. For each coordinate $(x, y)$ in $\hat{B}_{t-k}$, the corresponding velocity vector $M_{t-k}(x, y) = [v_x, v_y]$ is used to compute the positional shift $\Delta x = v_x \cdot t_s$ and $\Delta y = v_y \cdot t_s$, where $t_s$ denotes the duration of a single frame. These positional shifts are applied to the feature values if the velocity magnitude exceeds a predefined threshold $\tau_v$. The shifted feature maps are then obtained as

$$B'_{t-k}(x, y) = \frac{1}{|S(x, y)|} \sum_{(i,j) \in S(x,y)} \hat{B}_{t-k}(i, j) \tag{7}$$

where

$$S(x, y) = \{(i, j) : x = i + \lfloor \Delta x \rceil, y = j + \lfloor \Delta y \rceil, |M_{t-k}(i, j)| > \tau_v\}. \tag{8}$$

Here $|S(x, y)|$ denotes the cardinality of $S(x, y)$ and $\lfloor \cdot \rceil$ denotes the rounding operation. The shifted feature map $B'_{t-k}$ is then concatenated with the feature map $B_{t-k+1}$ at the next timestamp. To filter out any irrelevant features resulting from the shifting process, the concatenated feature map is element-wise multiplied with the occupancy score map $O_{t-k+1}$ as

$$\hat{B}_{t-k+1} = \text{concat}(B'_{t-k}, B_{t-k+1}) \odot O_{t-k+1}. \tag{9}$$

This process is iteratively applied $N$ times, generating the BEV feature maps $\hat{B}_{t-N+1}, \ldots, \hat{B}_t$ sequentially. The final $\hat{B}_t$ is then passed through a 1x1 convolution to obtain the final BEV feature map. This sequential nature of the process enhances overall comprehension of the MGTF module.

By incorporating motion information and occupancy score maps, the MGTF module captures the dynamics of moving objects while filtering out irrelevant features, resulting in a more robust BEV representation. Compared to traditional methods, MGTF captures trajectories of moving objects to enhance 3D object detection performance.

Table 1: Performance comparisons with 3D object detector on the nuScenes val set. 'L', 'C', and 'R' represent LiDAR, camera, and radar, respectively. †: trained with CBGS. ‡: use TTA.

| Method | Input | Backbone | Image Size | NDS | mAP | mATE | mASE | mAOE | mAVE | mAAE | FPS |
|---|---|---|---|---|---|---|---|---|---|---|---|
| CenterPoint-P† [23] | L | Pillars | - | 59.8 | 49.4 | 0.320 | 0.262 | 0.377 | 0.334 | 0.198 | - |
| CenterPoint-V† [23] | L | Voxel | - | 65.3 | 56.9 | 0.285 | 0.253 | 0.323 | 0.272 | 0.186 | - |
| CenterFusion† [3] | C+R | DLA34 | 448×800 | 45.3 | 33.2 | 0.649 | 0.263 | 0.535 | 0.540 | 0.142 | - |
| BEVDepth† [17] | C | R50 | 256×704 | 47.5 | 35.1 | 0.639 | 0.267 | 0.479 | 0.428 | 0.198 | 11.6 |
| RCBEV4d† [31] | C+R | R50 | 256×704 | 49.7 | 38.1 | 0.526 | 0.272 | 0.445 | 0.465 | 0.185 | - |
| CRAFT† [1] | C+R | DLA34 | 448×800 | 51.7 | 41.1 | 0.494 | 0.276 | 0.454 | 0.486 | 0.176 | 4.1 |
| RCM-Fusion† [4] | C+R | R50 | 450×800 | 52.8 | 44.4 | 0.527 | 0.272 | 0.450 | 0.515 | 0.180 | - |
| SOLOFusion† [20] | C | R50 | 256×704 | 53.4 | 42.7 | 0.567 | 0.274 | 0.411 | 0.252 | 0.188 | 11.4 |
| CRN [6] | C+R | R50 | 256×704 | 56.0 | 49.0 | 0.487 | 0.277 | 0.542 | 0.344 | 0.197 | 20.4 |
| RCBEVDet† [2] | C+R | R50 | 256×704 | 56.8 | 45.3 | 0.486 | 0.285 | 0.404 | 0.220 | 0.192 | 21.3 |
| CRT-Fusion-light† | C+R | R50 | 256×704 | 57.8 | 48.8 | 0.480 | 0.265 | 0.480 | 0.248 | 0.189 | 20.5 |
| CRT-Fusion | C+R | R50 | 256×704 | 57.2 | 50.0 | 0.499 | 0.277 | 0.531 | 0.261 | 0.192 | 14.5 |
| CRT-Fusion† | C+R | R50 | 256×704 | **59.7** | **50.8** | 0.461 | 0.264 | 0.419 | 0.234 | 0.186 | 14.5 |
| MVFusion† [32] | C+R | R101 | 900×1600 | 45.5 | 38.0 | 0.675 | 0.258 | 0.372 | 0.833 | 0.196 | - |
| BEVFormer [16] | C | R101 | 900×1600 | 51.7 | 41.6 | 0.673 | 0.274 | 0.372 | 0.394 | 0.198 | 1.7 |
| BEVDepth† [17] | C | R101 | 512×1408 | 53.5 | 41.2 | 0.565 | 0.266 | 0.358 | 0.331 | 0.190 | 5.0 |
| SOLOFusion [20] | C | R101 | 512×1408 | 54.4 | 47.2 | 0.518 | 0.275 | 0.604 | 0.310 | 0.210 | - |
| SOLOFusion† [20] | C | R101 | 512×1408 | 58.2 | 48.3 | 0.503 | 0.264 | 0.381 | 0.246 | 0.207 | - |
| CRN [6] | C+R | R101 | 512×1408 | 59.2 | 52.5 | 0.460 | 0.273 | 0.443 | 0.352 | 0.180 | 7.2 |
| CRN‡ [6] | C+R | R101 | 512×1408 | 60.7 | 54.5 | 0.445 | 0.268 | 0.425 | 0.332 | 0.180 | - |
| CRT-Fusion | C+R | R101 | 512×1408 | **62.1** | **55.4** | 0.425 | 0.264 | 0.433 | 0.237 | 0.193 | 4.9 |

# 4 Experiment

## 4.1 Experimental setup

**Dataset** We evaluated our proposed method using the nuScenes dataset [27], a popular public dataset for autonomous driving. This dataset consists of 1,000 scenes, divided into 700 scenes for training, 150 scenes for validation, and 150 scenes for testing. Each scene contains approximately 20 seconds of data. The nuScenes dataset provides comprehensive 360-degree coverage with data from six cameras, and five radars. Keyframes are annotated at a frequency of 2Hz, covering 10 object classes. We use the official evaluation metrics provided by the nuScenes benchmark, which are mean Average Precision (mAP) and nuScenes Detection Score (NDS).

**Implementation details** We adopted BEVDepth [17] as our baseline model. For a fair comparison with existing methods, we employed ResNet [28], and ConvNeXt [29] as backbone encoders in the camera branch. In the radar branch, we accumulated the past 6 radar sweeps to obtain the input point clouds and used PointPillars [30] with randomly initialized weights as the backbone network. Our proposed CRT-Fusion model performed temporal fusion using the BEV features from the past 6 frames. We also introduce CRT-Fusion-Light, a lightweight version of CRT-Fusion, where the 2D-CNN backbone is removed from the radar pipeline. CRT-Fusion-Light performs temporal fusion using the BEV features from the past 3 frames. Detailed values of hyperparameters including learning rate, optimizer, and data augmentation methods are provided in Appendix.

## 4.2 Comparison to the state of the art

Table 1 compares our proposed CRT-Fusion method with state-of-the-art 3D object detection methods on the nuScenes validation set. CRT-Fusion consistently outperforms existing radar-camera fusion models and camera-only models across various camera backbone configurations. With the ResNet-50 backbone, CRT-Fusion achieves an improvement of 12.2% in NDS and 15.7% in mAP compared to the baseline model BEVDepth [17]. Additionally, CRT-Fusion achieves 1.2% higher NDS and 1.0% higher mAP than the state-of-the-art model CRN under the same configurations without class-balanced grouping and sampling (CBGS) [37]. Furthermore, when CBGS is applied, our approach outperforms the current best model, RCBEVDet [2] by 2.9% in NDS and 5.5% in mAP. When using the ResNet-101 backbone, CRT-Fusion surpasses CRN by 1.4% in NDS and 0.9% in mAP without

Table 2: Performance comparisons with 3D object detector on the nuScenes test set. 'L', 'C', and 'R' represent LiDAR, camera, and radar, respectively. ‡: use Test Time Augmentation.

| Method | Input | Backbone | NDS | mAP | mATE | mASE | mAOE | mAVE | mAAE |
|---|---|---|---|---|---|---|---|---|---|
| PointPillars [30] | L | Pillars | 55.0 | 40.1 | 0.392 | 0.269 | 0.476 | 0.270 | 0.102 |
| CenterPoint [23] | L | Voxel | 67.3 | 60.3 | 0.262 | 0.239 | 0.361 | 0.288 | 0.136 |
| KPConvPillars [33] | R | Pillars | 13.9 | 4.9 | 0.823 | 0.428 | 0.607 | 2.081 | 1.000 |
| RadarDistill [34] | R | Pillars | 43.7 | 20.5 | 0.461 | 0.263 | 0.525 | 0.336 | 0.072 |
| CenterFusion [3] | C+R | DLA34 | 44.9 | 32.6 | 0.631 | 0.261 | 0.516 | 0.614 | 0.115 |
| RCBEV [31] | C+R | Swin-T | 48.6 | 40.6 | 0.484 | 0.257 | 0.587 | 0.702 | 0.140 |
| MVFusion [32] | C+R | V2-99 | 51.7 | 45.3 | 0.569 | 0.246 | 0.379 | 0.781 | 0.128 |
| CRAFT [1] | C+R | DLA34 | 52.3 | 41.1 | 0.467 | 0.268 | 0.456 | 0.519 | 0.114 |
| BEVFormer [16] | C | V2-99 | 56.9 | 48.1 | 0.582 | 0.256 | 0.375 | 0.378 | 0.126 |
| BEVDepth [17] | C | ConvNeXt-B | 60.9 | 52.0 | 0.445 | 0.243 | 0.352 | 0.347 | 0.127 |
| SOLOFusion [20] | C | ConvNeXt-B | 61.9 | 54.0 | 0.453 | 0.257 | 0.376 | 0.276 | 0.148 |
| CRN‡ [6] | C+R | ConvNeXt-B | 62.4 | 57.5 | 0.416 | 0.264 | 0.456 | 0.365 | 0.130 |
| SparseBEV [35] | C | V2-99 | 63.6 | 55.6 | 0.485 | 0.244 | 0.332 | 0.246 | 0.117 |
| StreamPETR [36] | C | V2-99 | 63.6 | 55.0 | 0.493 | 0.241 | 0.343 | 0.243 | 0.123 |
| RCBEVDet [2] | C+R | V2-99 | 63.9 | 55.0 | 0.390 | 0.234 | 0.362 | 0.259 | 0.113 |
| CRT-Fusion | C+R | ConvNeXt-B | 64.9 | 58.3 | 0.365 | 0.261 | 0.405 | 0.262 | 0.132 |
| CRT-Fusion‡ | C+R | ConvNeXt-B | **65.6** | **58.9** | 0.358 | 0.258 | 0.392 | 0.248 | 0.130 |

test time augmentation (TTA). Our lightweight version, CRT-Fusion-light, maintains a comparable FPS while delivering better performance compared to existing radar-camera 3D object detectors, demonstrating the efficiency and effectiveness of our approach.

Table 2 shows the performance of CRT-Fusion on the nuScenes test set. Our method outperforms all existing radar-camera fusion models, achieving state-of-the-art performance in both settings with and without TTA. Note that the V2-99 backbone is pre-trained on the external depth dataset DDAD [38].

## 4.3 Ablation studies

We conducted comprehensive ablation studies on the nuScenes validation set to evaluate the effectiveness of the key components in CRT-Fusion. Throughout these experiments, unless otherwise specified, we used a ResNet-50 backbone and an image size of $256 \times 704$ for the camera branch, and a BEV size of $128 \times 128$. All models are trained for 24 epochs without applying CBGS.

**Component analysis.** To assess the effect of each component, we gradually added each to our baseline model and analyze the performance improvements, as shown in Table 3. The first row shows the performance of the BEVDepth model as reported in the paper, achieving an NDS of 47.5% and an mAP of 35.1%. Our baseline model, represented in the second row, reproduces BEVDepth without CBGS and incorporates long-term temporal fusion [20], achieving an NDS of 47.4% and an mAP of 37.8%. By fusing radar and camera features at the BEV stage, including gating fusion, we observe a significant improvement, reaching an NDS of 55.4% and an mAP of 47.8%. RCA, which fuses radar features in the frustum view, contributes to an additional 1.2% and 1.1% improvement in NDS and mAP, respectively. Finally, by using MFE and MGTF, we achieve an additional gain of 1.1% in both NDS and mAP, resulting in the highest performance with an NDS of 57.2% and an mAP of 50.0%. These results demonstrate the effectiveness of each key component in CRT-Fusion, with our proposed modules providing significant improvements over the baseline model.

**Robustness to diverse weather and lighting conditions.** In Table 4, we analyze the performance of our model under varying weather and lighting conditions. For a fair comparison, we use the same settings as CRN, with a ResNet-101 backbone and an input size of $512 \times 1408$. The baseline model BEVDepth shows low mAP scores due to the impact of weather and light on camera sensors. However, by incorporating radar sensors, which are less affected by these factors, CRT-Fusion achieves over 15% higher mAP in all scenarios. Compared to CRN, the state-of-the-art camera-radar 3D object detection model, CRT-Fusion achieves higher mAP in all conditions except Sunny, offering particularly notable gains in night environments.

**Impact of accurate velocity estimation on MFE and MGTF.** Table 5 demonstrates the importance of accurate velocity prediction for each BEV grid. When the MFE and MGTF modules are applied to

Table 3: Ablation study of the main components of CRT-Fusion.

| Model Configuration | Input | NDS | mAP | mATE | mAOE |
|---------------------|-------|------|------|-------|-------|
| BEVDepth [17] | C | 47.5 | 35.1 | 0.639 | 0.479 |
| Baseline | C | 47.4 | 37.8 | 0.676 | 0.654 |
| + BEV Fusion | C+R | 55.4 | 47.8 | 0.528 | 0.584 |
| + RCA | C+R | 56.1 | 48.9 | 0.516 | 0.574 |
| + MFE & MGTF | C+R | **57.2** | **50.0** | **0.497** | **0.532** |

Table 4: Performance comparison under different weather and lighting conditions.

| Method | Input | Sunny | Rainy | Day | Night |
|--------|-------|-------|-------|------|-------|
| BEVDepth [17] | C | 39.0 | 39.0 | 39.3 | 16.8 |
| RCBEV [31] | C+R | 36.1 | 38.5 | 37.1 | 15.5 |
| RCM-Fusion [4] | C+R | 49.4 | 51.4 | 50.1 | 25.6 |
| CRN [6] | C+R | **54.8** | 57.0 | 55.1 | 30.4 |
| CRT-Fusion | C+R | 54.7 | **57.9** | **55.8** | **33.0** |

Table 5: Ablation study of the MFE and MGTF module applied to both the camera-based model ([17]) and our proposed model.

| Method | MFE & MGTF | NDS | mAP | mAVE |
|--------|-----------|------|------|-------|
| BEVDepth | X | 47.4 | 37.8 | 0.312 |
|          | O | 46.9 | 37.3 | 0.349 |
| CRT-Fusion | X | 56.1 | 48.9 | 0.278 |
|            | O | 57.2 | 50.0 | 0.265 |

Table 6: Comparison of radar-based view transformation methods. RGVT: Radar-Guided View Transformer. RVT: Radar-assisted View Transformation.

| Method | NDS | mAP | mATE | mAOE | mAVE |
|--------|------|------|-------|-------|-------|
| RGVT [39] | 48.6 | 41.3 | **0.571** | 0.603 | 0.522 |
| RVT [6] | 48.2 | 41.4 | 0.576 | 0.577 | 0.560 |
| RCA | **50.5** | **41.7** | 0.573 | **0.541** | **0.432** |

the camera-based baseline BEVDepth, performance degrades, as shown in the second row of the table. The NDS and mAP scores both drop by 0.5%, highlighting the challenge of accurately estimating velocities solely from camera information. In contrast, CRT-Fusion leverages radar information to achieve more precise velocity predictions. By incorporating the MFE and MGTF modules, CRT-Fusion achieves an improvement of 1.1% in both NDS and mAP, demonstrating the effectiveness of these modules in enhancing the performance. These results indicate that the successful operation of the MFE and MGTF modules is highly dependent on accurate velocity estimation. In the absence of radar data, velocity estimation accuracy is significantly compromised, suggesting that MFE and MGTF modules are optimally suited for radar-camera fusion frameworks.

**Comparison of radar-based view transformation methods.** Table 6 compares various view transformation methods leveraging radar information. For a fair comparison, we used CRN as the baseline model and conducted experiments in a single-frame setting. The LSS approach predicts 3D depth from camera features and transforms them into BEV features, but can generate inaccuracies due to imprecise depth information. RGVT [39] projects radar points onto the image plane and encodes radar depth features using a lightweight CNN, which are then combined with image features to predict 3D depth. RVT [6] refines the depth distribution predicted from camera features using radar occupancy information. Our proposed RCA outperforms existing methods in almost all metrics, demonstrating its effectiveness in utilizing radar information to transform camera features into accurate BEV features. The superior performance of RCA demonstrates that leveraging radar information for depth prediction significantly enhances BEV perception.

**Qualitative results.** Figure 4 presents a qualitative comparison between our proposed CRT-Fusion model and the previous state-of-the-art CRN model across various real-world scenarios. CRT-Fusion demonstrates enhanced detection capabilities, accurately identifying objects and showing superior precision in predicting orientations and center positions. Furthermore, CRT-Fusion maintains high accuracy across diverse conditions, effectively capturing multiple targets. These results underscore the robustness and enhanced spatial perception of CRT-Fusion. Additional qualitative results are available in the Supplemental Materials.

## 5 Discussion and Conclusion

**Limitations and future work.** While CRT-Fusion achieves significant performance gains over the baseline, the computational cost increases with the number of previous frames used for temporal fusion, limiting the number of frames that can be incorporated due to hardware constraints. This

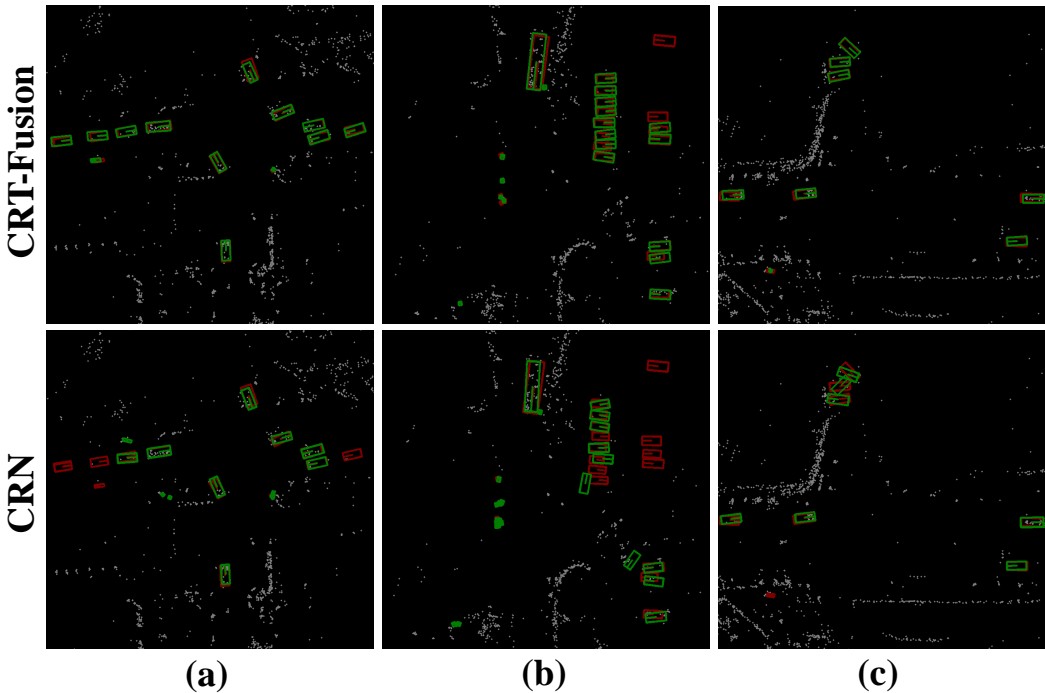

**CRT-Fusion**

**CRN**

**(a)**       **(b)**       **(c)**

Figure 4: **Qualitative results comparing CRT-Fusion and CRN:** Green boxes indicate CRT-Fusion prediction boxes, blue boxes denote CRN prediction boxes, and red boxes represent ground truth (GT) boxes.

issue is likely due to the parallel fusion structure used for combining BEV features. To address this, a potential solution is to adopt a recurrent fusion structure, which fuses BEV features temporally in a sequential manner. This approach could maintain computational feasibility while incorporating long-term historical BEV features. Future work will explore this recurrent fusion architecture for CRT-Fusion to further reduce its computational complexity.

**Conclusion.** In this paper, we introduced CRT-Fusion, a novel framework that integrates temporal information into radar-camera fusion for 3D object detection. By explicitly taking the motion of dynamic objects into account through our proposed Motion Feature Estimator and Motion Guided Temporal Fusion modules, CRT-Fusion significantly improves detection accuracy and robustness in complex real-world scenarios. Our Multi-View Fusion module enhances depth prediction by leveraging radar features to improve image features before fusing them into a unified BEV representation. Extensive experiments on the challenging nuScenes dataset demonstrate that CRT-Fusion achieves state-of-the-art performance in the radar-camera-based 3D object detection category, surpassing all existing methods. Additionally, our approach demonstrates remarkable robustness under diverse weather and lighting conditions, highlighting its potential for real-world deployment. We believe that our work will inspire further research on the fusion of temporal and multi-modal information for robust perception in adverse environments.

## Acknowledgement

This work was partly supported by the Institute of Information & Communications Technology Planning & Evaluation (IITP) grant funded by the Korea government (MSIT) (No. 2022-0-00957, Distributed on-chip memory-processor model PIM (Processor in Memory) semiconductor technology development for edge applications); the IITP grant funded by the Korea government (MSIT) (No. RS-2021-II211343, Artificial Intelligence Graduate School Program, Seoul National University); and the National Research Foundation of Korea (NRF) grant funded by the Korea government (MSIT) (No. 2020R1A2C2012146).

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

# Supplementary Materials for CRT-Fusion

In this Supplementary Material, we provide additional details that were not covered in the main paper. We organize the content as follows: comprehensive formulation of loss functions (Section A), architectural specifications and training protocols (Section B), extensive ablation studies and computational efficiency analysis (Section C), in-depth qualitative evaluation on the nuScenes dataset (Section D), and discussion of societal implications (Section E).

## A   Loss Function

The total loss function used in CRF-Fusion is composed of five components: a standard 3D object detection loss and four additional losses derived from different head networks within our model. The total loss $L_{total}$ is given by

$$L_{total} = L_{det} + \lambda_{depth}L_{depth} + \lambda_{seg}L_{seg} + \lambda_{vel}L_{vel} + \lambda_{occ}L_{occ}, \tag{10}$$

where $L_{det}$ is the 3D object detection loss, $L_{depth}$ is the loss from the Depth Prediction Head in MVF, $L_{seg}$ is the loss from the Perspective-View Semantic Segmentation Head in MVF, $L_{vel}$ is the loss from the Velocity Prediction Head in MFE, and $L_{occ}$ is the loss from the Object Occupancy Prediction Head in MFE. The parameters $\lambda_{depth}$, $\lambda_{seg}$, $\lambda_{vel}$, and $\lambda_{occ}$ are the weights for the corresponding loss terms. The Depth Prediction Loss uses binary cross-entropy loss for depth estimation, with a weight of $\lambda_{depth} = 3.0$, following the approach used in BEVDepth. For the Perspective View Segmentation Loss, we also employ the binary cross-entropy loss, with a weight of $\lambda_{seg} = 25$, inspired by SA-BEV. The Velocity Prediction Loss, which handles velocity $(v_x, v_y)$ and orientation prediction, utilizes Mean Squared Error (MSE) with a weight of $\lambda_{vel} = 1$. Finally, the BEV Object Occupancy Loss uses Binary Focal Loss for foreground and background segmentation, with a weight of $\lambda_{occ} = 30$.

## B   Implementation Details

The nuScenes datasets [27] are publicly available to use under CC BY-NC-SA 4.0 license and can be downloaded from `https://www.nuscenes.org/`. We implemented our model using the MMDetection3D [40] codebase and trained it for a total of 24 epochs. The training process consists of two phases. In the initial phase, the model is trained for 6 epochs without the MGTF module. Then, the entire model is trained for the remaining 18 epochs. For the ResNet50-based model, we used 4 NVIDIA RTX 3090 GPUs for training, while for ResNet101 and ConvNeXt-B, we used 3 NVIDIA A100 GPUs. Table 7 summarizes the training settings for different camera backbone networks.

In the perspective view, we apply data augmentation techniques consistent with previous studies [41, 17, 6], including horizontal random flipping, random scaling ($[-0.06, 0.11]$), and random rotation ($\pm 0.54°$). For the bird's-eye view, we employ random flipping along the x and y axes, random scaling ($[0.95, 1.05]$), and random rotation ($\pm 0.3925$ rad). Additionally, we use a technique that randomly drops radar sweeps and points [42]. To ensure a fair comparison with other models, we do not utilize ground-truth sampling augmentation (GT-AUG) [43], which is commonly used in LiDAR-based models.

## C   Additional Experimental Results

### C.1   Analysis of the Motion Feature Estimation Module.

The Motion Feature Estimation (MFE) module generates a motion-aware BEV feature map by estimating velocity for each grid and using it to refine the BEV representation. While MFE can be integrated into various architectures, its effectiveness is highly reliant on accurate velocity predictions. To demonstrate this, we applied MFE to both the camera-based model BEVDepth and our proposed CRT-Fusion, which incorporates radar data.

Figure 5 shows the results of applying the MFE module to each model. The red boxes represent Ground Truth (GT) boxes, the red arrows indicate GT velocity, and the white arrows show the predicted velocity from the MFE module. The size and direction of the arrows reflect the velocity's

Table 7: Training settings for different backbone networks.

| Configs | ResNet-50 | ResNet-101 | ConvNext-B |
|---|---|---|---|
| Image size | $256 \times 704$ | $512 \times 1408$ | $512 \times 1408$ |
| BEV size | $128 \times 128$ | $256 \times 256$ | $256 \times 256$ |
| Optimizer | AdamW | AdamW | AdamW |
| Base Learning Rate | 2e-4 | 1e-4 | 1e-4 |
| Weight Decay | 1e-7 | 1e-7 | 1e-2 |
| Optimizer Momentum | 0.9, 0.999 | 0.9, 0.999 | 0.9, 0.999 |
| Batch Size | 16 | 12 | 12 |
| Training Epochs | 24 | 24 | 24 |
| LR Schedule | Step Decay | Step Decay | Step Decay |
| Gradient Clip | 5 | 5 | 5 |

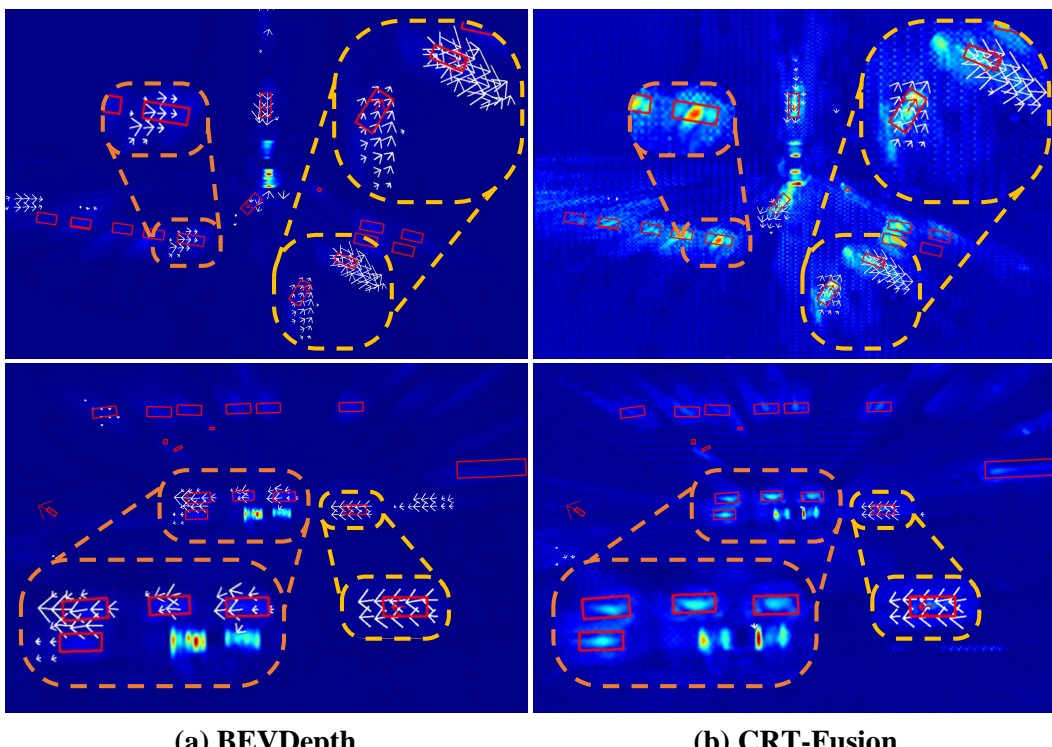

(a) BEVDepth      (b) CRT-Fusion

Figure 5: **Comparison of velocity prediction using the MFE module in BEVDepth and CRT-Fusion.** Red boxes are the Ground Truth (GT) boxes, red arrows show GT velocity, and white arrows represent predicted velocity. Yellow highlights indicate areas where CRT-Fusion predicts velocity more accurately, while orange highlights show static objects correctly identified by CRT-Fusion but misclassified by BEVDepth.

magnitude and direction. The yellow and orange highlights illustrate the differences in velocity prediction accuracy and the ability to distinguish static objects. Specifically, in the yellow-highlighted areas, CRT-Fusion predicts velocities closely aligned with GT, whereas BEVDepth shows inaccuracies. In the orange-highlighted areas, CRT-Fusion accurately identifies static objects, while BEVDepth misclassifies them as dynamic. In conclusion, the effectiveness of the MFE module is closely tied to the accuracy of velocity estimation. Integrating radar data in CRT-Fusion significantly enhances the module's ability to generate reliable motion-aware BEV features, underscoring the value of sensor fusion for robust 3D object detection.

Table 8: Ablation study of temporal frames.

| # of prev frames | NDS | mAP | mATE | mAOE | mAVE |
|---|---|---|---|---|---|
| 0 | 50.4 | 44.4 | 0.515 | 0.598 | 0.567 |
| 1 | 55.1 | 47.1 | 0.508 | 0.570 | 0.295 |
| 2 | 56.0 | 48.1 | 0.506 | 0.542 | 0.280 |
| 3 | 56.6 | 48.8 | 0.498 | 0.540 | 0.269 |
| 4 | 56.7 | 49.3 | 0.505 | 0.556 | 0.268 |
| 5 | 56.6 | 49.2 | 0.506 | 0.557 | 0.262 |
| 6 | 57.2 | **50.0** | 0.494 | 0.557 | 0.265 |
| 7 | **57.3** | 49.8 | 0.497 | 0.532 | 0.265 |
| 8 | **57.3** | 49.7 | 0.499 | 0.531 | 0.261 |

Table 9: Ablation study of the hyperparameters of CRT-Fusion.

| $\tau_P$ | NDS | mAP |
|---|---|---|
| 0.15 | 56.8 | 49.4 |
| 0.20 | 56.6 | 49.0 |
| 0.25 | **57.2** | **50.0** |
| 0.30 | 56.7 | 49.4 |

(a) $\tau_P$: Perspective view segmentation threshold

| $\tau_B$ | NDS | mAP |
|---|---|---|
| 0.00 | 56.7 | 49.2 |
| 0.05 | **57.2** | **50.0** |
| 0.10 | 56.0 | 48.7 |
| 0.15 | 55.8 | 48.4 |

(b) $\tau_B$: Bird eye's view segmentation threshold

| $\tau_v$ | NDS | mAP |
|---|---|---|
| 0.0 | 57.0 | 49.2 |
| 0.5 | 56.6 | 48.9 |
| 1.0 | **57.2** | **50.0** |
| 1.5 | 56.8 | 49.3 |

(c) $\tau_v$: Motion estimation threshold

| # of grid | NDS | mAP |
|---|---|---|
| 32 | 56.1 | 48.8 |
| 64 | 56.8 | 49.3 |
| 128 | **57.2** | **50.0** |
| 256 | 56.7 | 49.4 |

(d) Number of radar grid in RCA

## C.2 Hyperparameter Analysis.

The ablation studies on the hyperparameters used in CRT-Fusion are shown in Tables 8 and 9. Table 8 examines the optimal number of previous frames to consider for achieving the best performance. The results show that the performance of CRT-Fusion improves as the number of frames increases, with 6 frames yielding the best results when considering computational cost and performance. Tables 9 (a) and (b) investigate the optimal segmentation thresholds for the perspective view and bird's eye view (BEV), respectively. In the perspective view, a segmentation threshold $\tau_P$ of 0.25 achieves the best performance, while in the BEV, a threshold $\tau_B$ of 0.05 yields the highest performance. Increasing the threshold may lead to the removal of foreground regions, resulting in a decline in performance. Table 9 (c) explores the velocity threshold $\tau_v$ for considering a BEV grid as a dynamic object in the MFE module. The results demonstrate that considering BEV grids with velocities above 1 m/s achieves the highest performance. Finally, Table 9 (d) examines the number of radar BEV grids to match with each image feature pixel in the RCA module. The experiments reveal that matching 128 grids yields the best performance.

## C.3 Evaluating Model Efficiency.

Table 10 presents the analysis of performance, latency, and GPU memory usage of CRT-Fusion and CRN as the number of past frames increases. We reproduced the CRN model following the official implementation. For a fair comparison, both models employ identical camera and radar backbones with consistent input image dimensions. All experimental evaluations were conducted on a single NVIDIA RTX 3090 GPU and an Intel Xeon Silver 4210R CPU.

Our experimental results demonstrate that CRT-Fusion consistently achieves superior performance in both NDS and mAP metrics across all temporal configurations. Notably, as we extend the temporal context by incorporating additional past frames, CRT-Fusion exhibits stable resource usage with minimal degradation. The memory efficiency of our approach is evidenced by its peak consumption of 3.754 GB at 7 frames, while CRN reaches 4.342 GB under identical conditions. In terms of computational latency, CRT-Fusion demonstrates robust scalability, with inference time increasing

Table 10: Quantitative results comparing of CRT-Fusion and CRN. Comparison of accuracy (NDS, mAP) and efficiency (GPU memory, Latency) of CRT-Fusion and CRN with increasing number of history frames. Mem. represents the GPU memory consumption at inference phase.

| # of prev frames | CRT-Fusion | | | | CRN | | | |
|---|---|---|---|---|---|---|---|---|
| | NDS | mAP | Mem (GB) | Latency (ms) | NDS | mAP | Mem (GB) | Latency (ms) |
| 0 | 50.35 | 44.37 | 3.686 | 57.1 | 44.24 | 41.36 | 4.232 | 55.6 |
| 1 | 55.08 | 47.07 | 3.692 | 62.0 | 53.14 | 43.80 | 4.244 | 85.0 |
| 2 | 56.03 | 48.07 | 3.698 | 62.8 | 54.79 | 45.88 | 4.270 | 120.2 |
| 3 | 56.60 | 48.82 | 3.706 | 64.2 | 55.57 | 46.87 | 4.286 | 138.6 |
| 4 | 56.66 | 49.29 | 3.714 | 64.9 | 56.01 | 47.10 | 4.302 | 187.1 |
| 5 | 56.55 | 49.19 | 3.724 | 66.2 | 55.83 | 47.18 | 4.310 | 212.0 |
| 6 | 57.15 | 50.01 | 3.744 | 67.1 | 55.55 | 47.40 | 4.326 | 243.5 |
| 7 | 57.30 | 49.75 | 3.754 | 69.6 | 56.30 | 47.61 | 4.342 | 273.0 |

moderately from 57.1 ms to 69.6 ms when expanding from 0 to 7 frames. CRN also benefits from increased temporal information, although it experiences a more substantial increase in computational overhead, with latency reaching 273 ms at 7 frames. These empirical results highlight the efficiency of our proposed architecture in maintaining real-time inference capabilities while utilizing temporal information effectively.

### C.4 Latency Analysis of CRT-Fusion and CRT-Fusion-light.

Table 11 presents the component-wise latency analysis of our CRT-Fusion variants. CRT-Fusion-light, which incorporates a lightweight radar backbone and processes fewer temporal frames, achieves an overall latency of 48.8 ms compared to 67.1 ms of the original architecture. The efficiency gains primarily originate from the radar processing pipeline, where the radar backbone latency decreases from 13.9 ms to 3.9 ms, and the MGTF module reduces from 15.2 ms to 8.6 ms.

Table 11: Ablation study of Inference Time.

| Methods | C.B. | R.B. | MVF | MFE | MGTF | Head | Total |
|---|---|---|---|---|---|---|---|
| CRT-Fusion | 13.4 ms | 13.9 ms | 15.0 ms | 0.7 ms | 15.2 ms | 8.9 ms | 67.1 ms |
| CRT-Fusion-light | 13.4 ms | 3.9 ms | 13.3 ms | 0.7 ms | 8.6 ms | 8.9 ms | 48.8 ms |

## D   Qualitative results of CRT-Fusion.

Figure 6 showcases qualitative results of our proposed CRT-Fusion method on the nuScenes validation set. We compare the object detection performance of CRT-Fusion with the baseline model, BEVDepth, by visualizing the predicted bounding boxes in the BEV representation. Both models employ a ResNet-101 as the camera backbone for feature extraction. The examples demonstrate the effectiveness of CRT-Fusion in various driving scenarios, such as urban streets, intersections, and highways. Our model consistently produces more accurate and well-aligned bounding boxes compared to the baseline model.

## E   Discussions of potential societal impacts.

CRT-Fusion has the potential to enhance the accuracy and robustness of 3D object detection in autonomous vehicles and robotics systems by fusing radar, camera, and temporal information. Despite its benefits, the reliance on sophisticated technology and data fusion could lead to increased costs and complexity, potentially limiting accessibility and widespread adoption.

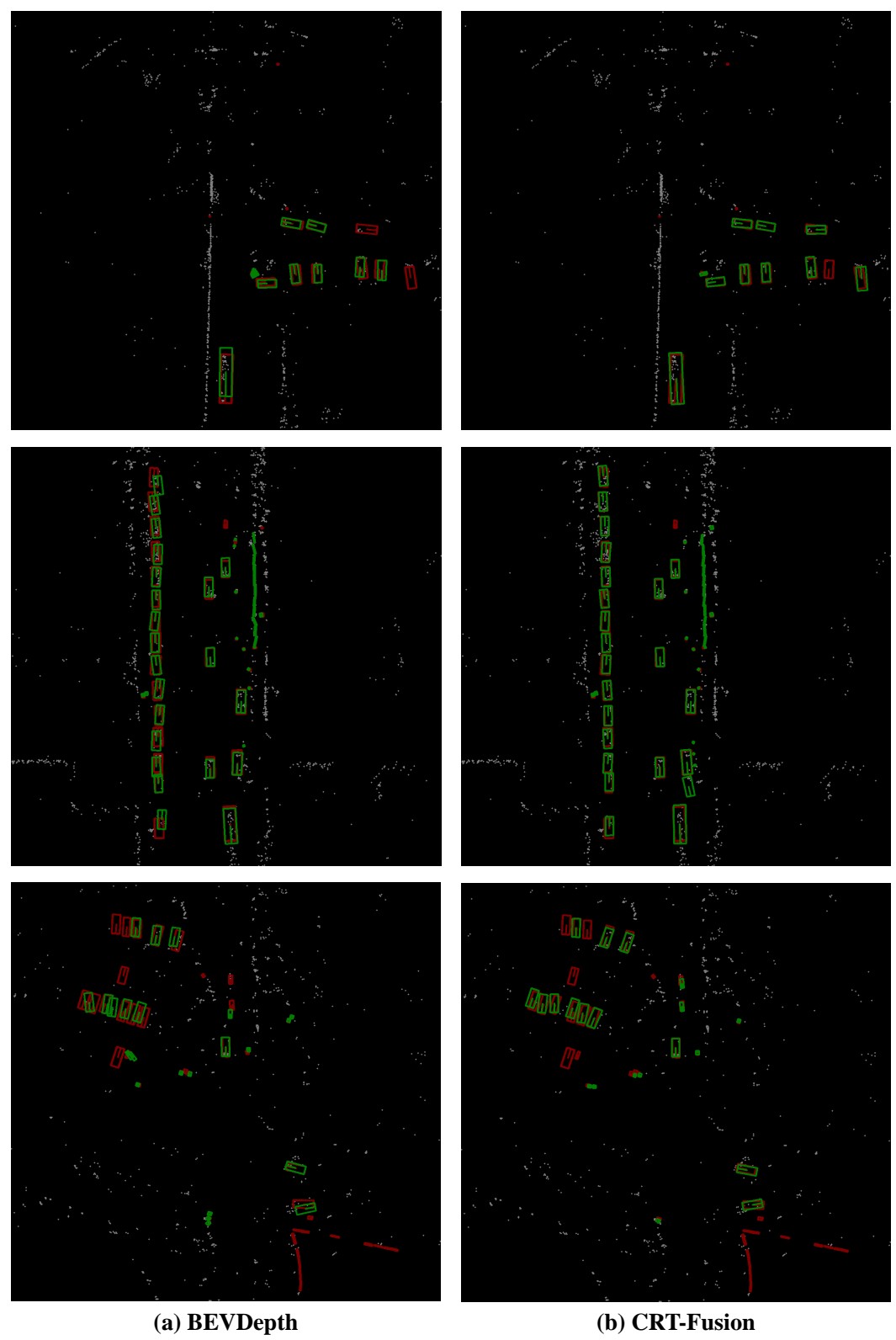

**(a) BEVDepth**  **(b) CRT-Fusion**

Figure 6: **Qualitative results under different scenarios on the nuScenes validation set.** Red boxes represent ground truth annotations, while blue and green boxes indicate the predicted bounding boxes from BEVDepth and CRT-Fusion, respectively. The white points represent the radar point cloud.

