# OpenReview forum: "CRT-Fusion: Camera, Radar, Temporal Fusion Using  Motion Information  for 3D Object Detection"
_NeurIPS.cc/2024/Conference — NeurIPS 2024 poster_

### Official Review · Reviewer_mr4N · 2024-06-23

**Soundness:** 3
**Presentation:** 3
**Contribution:** 3
**Rating:** 6
**Confidence:** 5

**Summary:**

This paper proposes a radar-camera and temporal fusion method (CRT) for the 3D object detection task. CRT design Multi-View Fusion (MVF), Motion Feature Estimator (MFE), and Motion Guided Temporal Fusion (MGTF) modules. MVF employs radar information for 2D-to-3D projection. MGTF utilizes velocity and occupancy predictions from MFE to fuse multi-frame BEV features. The proposed method achieves state-of-the-art 3D object detection results on nuScenes.

**Strengths:**

1. The MGTF module is interesting in decoupling and fusing multi-frame object features on dense BEV features.
2. The paper achieves a good balance between performance and computational efficiency, making it a promising advancement for real-time radar processing in autonomous driving.
3. The paper achieves a SOTA for radar-camera 3D object detection

**Weaknesses:**

1. The proposed method requires velocity supervision, while some datasets, such as Waymo, do not provide velocity labels.
2. Table 1 shows that the proposed has well speed-accuracy trade-offs. It is better to provide the inference time of each part in the model.
3. It seems that hyperparameters have a significant impact on model accuracy (Table 9).
4. The paper does not show the model generalization of the proposed method. It is only applied to BEVDepth.
5. BEV segmentation in MFE is misleading. It is better to use terms like object occupancy or foreground segmentation.

**Questions:**

1. Training details. How is the model trained? Is the multi-frame multi-modal model trained in one step?
2. What about the training cost?
3. What TTA is used in Table 2?
4. In Line 153, a set of M radar features are associated with W. What M does the model use? What if radar features are less than M?
5. What is B_{t−k}(x, y) in eq.4？Please give more details for object occupancy ground truth generation.
6. What do O and X mean in Table 5?

**Limitations:**

The authors address the limitations.

---

> ### Author Rebuttal · Authors · 2024-08-06
>
> **W1: The proposed method requires velocity supervision, while some datasets, such as Waymo, do not provide velocity labels.**
>
> The velocity ground truth (GT) for objects is provided as state information in the nuScenes dataset. Without this velocity information, we may need to calculate the derivatives from the object tracks over multiple frames.
>
> **W2: Table 1 shows that the proposed has well speed-accuracy trade-offs. It is better to provide the inference time of each part in the model.**
>
> As per the reviewer's suggestion, we provide the breakdown of the inference time for CRT-Fusion and CRT-Fusion-light below
>
> |                    | Camera Backbone | Radar Backbone |  MVF  |  MFE  | MGTF | Detection Head |  Total  |
> |--------------------|:---------------:|:--------------:|:-----:|:-----:|:----:|:-----------------:|:-------:|
> | CRT-Fusion         |     13.4 ms     |    13.9 ms     | 15 ms | 0.7 ms|15.2 ms|      8.9 ms       | 67.1 ms |
> | CRT-Fusion-light   |     13.4 ms     |     3.9 ms     |13.3 ms| 0.7 ms| 8.6 ms|      8.9 ms       | 48.8 ms |
>
> CRT-Fusion-light uses a radar backbone network with lower capacity and combines fewer past frames compared to CRT-Fusion. Therefore, the main differences in inference time are due to the radar backbone, MVF, and MGTF components.
>
> **W3: It seems that hyperparameters have a significant impact on model accuracy (Table 9).**
>
> The hyperparameters might impact our model's accuracy, requiring tuning to optimize performance. This is a common behavior observed in other 3D object detection models.
>
> **W4: The paper does not show the model generalization of the proposed method. It is only applied to BEVDepth.**
>
> Many BEV-based 3D object detectors employ the LSS structure presented in BEVDepth. Therefore, our method is applicable to any BEV frameworks employing LSS. We have not yet applied our method to Transformer-based BEV frameworks such as BEVFormer, which will be explored in future work.
>
> **W5: BEV segmentation in MFE is misleading. It is better to use terms like object occupancy or foreground segmentation.**
>
> As the reviewer suggested, we will rename this as "object occupancy" to avoid confusion in the final revision of the paper.
>
> **Q1: Training details. How is the model trained? Is the multi-frame multi-modal model trained in one step?**
>
> As detailed in Supplementary A, we trained the model using single-frame images for the initial 6 epochs, and then included temporal fusion for the subsequent 18 epochs.
>
> **Q2: What about the training cost?**
>
> Training times without using Class Balanced Grouping and Sampling (CBGS) are provided below:
>
> ResNet 50: 15 hours on 4 3090 GPUs
>
> ResNet101: 26 hours on 3 NVIDIA A100 GPUs
>
> ConvNeXt-B: 35 hours on 3 NVIDIA A100 GPUs
>
> **Q3: What TTA is used in Table 2?**
>
> We applied TTA using  flipping in both BEV and image domains.
>
> **Q4: In Line 153, a set of $M$ radar features are associated with $W$. What $M$ does the model use? What if radar features are less than $M$?**
>
> We use the setting $M=128$, which was determined empirically. If there are fewer radar points to cluster than $M$, we pad with zeros. Our ablation study  for $M$ are provided in Table 9 (d) of the Supplementary Materials.
>
> **Q5: What is $B_{t-k}(x, y)$ in eq. 4？ Please give more details for object occupancy ground truth generation.**
>
> We apologize for any confusion regarding the calculation of GT for MFE. It is confusing to use $B_{t-k}(x,y)$ in Equation 4.
>
> In Equation 4, $r(x,y)$ represents the Intersection over Union (IoU) ratio between the physical box corresponding to the pixel at a position $(x,y)$ and the 3D object boxes projected onto the BEV domain. Pixels with an IoU ratio exceeding $\tau_{iou}$ are classified as positive and are assigned the GT velocity and GT occupancy state for supervision. We revised the equation 4 by replacing $B_{t-k}(x,y)$ with $H(x,y)$:
>
> Define the IoU ratio $r(x, y)$ for pixel $(x, y)$ as
>
>
> $r(x, y) = \frac{|H(x, y) \cap \mathcal{P}(\mathcal{G})|}{|H(x, y) \cup \mathcal{P}(\mathcal{G}) |}
> $
>
> where \( H(x, y) \) is the physical bounding box whose size corresponds to the pixel at $(x, y)$, $\mathcal{G}$ is the set of 3D object boxes, and $\mathcal{P}(\mathcal{G})$ is the projection of these boxes onto the BEV domain.
>
> **Q6: What do O and X mean in Table 5?**
>
> In Table 5 of the manuscript, 'O' indicates the use of the MFE and MGTF modules, while 'X' indicates non-use. Some mistakes in Table 5 have been corrected below
>
> | Method      | MFE & MGTF | NDS  | mAP  | mAVE  |
> |-------------|:----------:|:----:|:----:|:-----:|
> | BEVDepth    |      X     | 47.4 | 37.8 | 0.312 |
> | BEVDepth    |     O     | 46.9 | 37.3 | 0.349 |
> | CRT-Fusion  |      X     | 56.1 | 48.9 | 0.278 |
> | CRT-Fusion  |     O     | 57.2 | 50.0 | 0.265 |

---

> > ### Comment · Reviewer_mr4N · 2024-08-08
> >
> > The rebuttal has addressed most of my concerns. However, I still have a question about the object occupancy ground truth generation (Q5). What does the physical bounding box mean? Besides, in the corrected Table 5 (Q6), it seems using MFE and MGTF modules for BEVDepth decreases the performance.

---

> > > ### Author Response · Authors · 2024-08-08
> > >
> > > **What does the physical bounding box mean?**
> > >
> > > The term "physical bounding box" refers to an anchor box whose size in the real world corresponds to that of a pixel. During our encoding process, actual scenes in the physical world shrink to features of smaller size. Thus, the actual size of a single pixel corresponds to a 2D box of $0.8m \times 0.8m$ in the x and y axes of the BEV  domain. We computed the IoU (Intersection over Union) ratio between this anchor box and the 2D ground truth (GT) box projected onto the BEV to determine whether a pixel is positive.
> > >
> > >
> > > **Besides, in the corrected Table 5 (Q6), it seems using MFE and MGTF modules for BEVDepth decreases the performance.**
> > >
> > > In this experiment, we applied MFE and MGTF to BEVDepth, which does not utilize radar data for BEV perception. While CRT-Fusion employs MFE and MGTF to enhance fused BEV features derived from both radar and camera inputs, we applied these modules solely to the camera-derived BEV features of BEVDepth.
> > >
> > > Our results indicate that MFE and MGTF do not enhance performance in the absence of radar data. This behavior demonstrates that the information provided by radar data is crucial for MFE and MGTF to achieve performance gains over the baseline.

---

> > > > ### Comment · Reviewer_mr4N · 2024-08-09
> > > >
> > > > The authors successfully address all my questions. I would like to keep my original rating and am inclined to accept this paper.

---

### Official Review · Reviewer_EC1R · 2024-07-04

**Soundness:** 3
**Presentation:** 3
**Contribution:** 3
**Rating:** 5
**Confidence:** 4

**Summary:**

This paper presents a method, called CRT-Fusion, for 3D object detection that fuses temporal information with radar-camera features. The CRT-Fusion captures the object motion with three modules: Multi-View Fusion (MVF), Motion Feature Estimator (MFE), and Motion Guided Temporal Fusion (MGTF). The MVF module uses Radar-Camera Azimuth Attention (RCA) to enhance the depth information in camera BEV features and combines them with radar features through a gated fusion network. The MFE module estimates pixel-wise velocity and performs BEV segmentation, which guides the temporal feature alignment and fusion in the MGTF module across multiple timestamps. The evaluation on the nuScenes dataset indicates that CRT-Fusion achieves state-of-the-art performance.

**Strengths:**

This paper is well-written and easy to follow. While there is prior work on fusing temporal and camera-radar features to improve 3D detection, the presented method appears novel as it learns how to estimate object pixel-wise velocity and occupancy to guide the fusion of object-level temporal features. The proposed Radar-Camera Azimuth Attention (RCA) is shown to effectively enhance image features with depth information from radar features. The experiments are well-designed, and the results on the nuScenes dataset show a significant performance gain over SOTA.

**Weaknesses:**

The paper could improve clarity in the following:
1.	As mentioned in lines 181-185, CRT-Fusion uses object-level pixel-wise velocity and occupancy to integrate temporal information with BEV features. CRT-Fusion's performance relies on the accuracy of both detection and velocity estimation. It is not clear how to address error propagation across frames.

2.	Regarding inference latency, some approaches (e.g., CRN[6]) use only 2 adjacent frames as input. According to the results in Table 8 of the supplementary material, the 2-frame version of CRT-Fusion performs similarly to (or worse than) CRN in terms of NDS and mAP metrics. This raises concerns that the performance gains of CRT-Fusion may be a trade-off involving longer temporal cues, leading to increased latency and complexity. The paper should discuss this trade-off.

3.	The batch size used during inference is not specified; Table 7 only presents the batch size used during training. Additionally, it appears that the authors directly used speed results from other papers for comparison, which may not be meaningful as the machines used for different models can vary significantly (Unfortunately, some other papers also directly cite and compare the speed) .

**Questions:**

In line 212, should it be $M_{t-k}(x, y)$? In the inference, the model does not have access to ground truth.

**Limitations:**

The paper poorly discusses limitations. Failure cases and potential improvements of the approach in the future should be discussed.

---

> ### Author Rebuttal · Authors · 2024-08-06
>
> **W1: It is unclear how to address error propagation across frames.**
>
> Error propagation in CRT-Fusion has been addressed by (1) taking a weighted sum of aligned BEV feature maps through the Gated Fusion Network (GFN) and (2) applying MGTF only for absolute velocity predictions above a threshold ($\tau_v$). The gating mechanism of GFN allows the model to reduce the contribution of misaligned features during feature aggregation. Additionally, velocity thresholding enables the model to be less affected by noisy velocity predictions. We will include these points in the final version.
>
> **W2: This raises concerns that the performance gains of CRT-Fusion may be a trade-off involving longer temporal cues, leading to increased latency and complexity.**
>
> To address the review's concern, we evaluated the performance, latency, and GPU memory usage of both CRT-Fusion and CRN as the number of past frames increases.  Figure 1 in the attached PDF file shows that our model consistently outperforms CRN in terms of NDS and mAP across all frame settings. This confirms that the performance gains are not solely attributed to using longer temporal information. Furthermore, CRT-Fusion demonstrates superior efficiency in terms of GPU memory usage and latency compared to CRN.
>
> **W3: Inference batch size is not specified.**
>
> We used a batch size of 1 for inference.
>
> **Q1: In line 212, should it be $M_{t-k}(x, y)$?**
>
> We thank the reviewer for finding the typo.  $M_{t-k}^{GT}(x, y)$ should be replaced with $M_{t-k}(x, y)$, which will be corrected in the final version of the paper.
>
> **L1: The analysis of limitations is insufficient.**
>
> Here is our discussion about the limitations of our study, which will be added in the final version:
> While our model has achieved significant performance gains over the baseline, the computation time increases with the number of previous frames used for temporal fusion. This limitation means our method cannot accommodate as many previous frames as desired due to hardware constraints. This issue might stem from the parallel fusion structure used to combine BEV features. One potential remedy is to adopt a recurrent fusion structure that combines BEV features temporally in a recurrent fashion. By doing so, the computational feasibility can be maintained while incorporating BEV features from long-term horizon. In the future, we will explore the recurrent fusion architecture for temporal fusion.

---

> > ### Comment · Reviewer_EC1R · 2024-08-07
> > **The rebuttal successfully addresses my review comments**
> >
> > The rebuttal successfully addresses my review comments. The rebuttal presents additional results that address my concerns about the latency of the proposed method.  I would like to keep my original rating.

---

### Official Review · Reviewer_dkEf · 2024-07-09

**Soundness:** 3
**Presentation:** 3
**Contribution:** 3
**Rating:** 4
**Confidence:** 4

**Summary:**

In this study, the authors present CRT-Fusion, an innovative framework designed to incorporate temporal information into radar-camera fusion, thereby enhancing the robustness of 3D object detection. The CRT-Fusion framework is composed of three integral modules: Multi-View Fusion (MVF), Motion Feature Estimator (MFE), and Motion Guided Temporal Fusion (MGTF).

**Strengths:**

This paper presents a novel framework capable of integrating temporal information into radar-camera fusion for enhanced 3D object detection.

Overall, the paper is well-written and clearly articulated. Experimental results demonstrate that this method outperforms previous approaches in 3D object detection.

**Weaknesses:**

Please consult the question section for further information.

**Questions:**

1. The paper lacks a detailed description of the loss function used in the proposed framework.

2. The authors claim that the Multi-View Fusion (MVF) module enhances depth prediction. However, no experimental evidence is provided to substantiate this assertion.

3. The Motion Feature Estimator (MFE) module is responsible for the pixel-wise velocity estimation task, yet the paper does not clarify how the velocity estimation is supervised during the learning process.

4. The overall improvement achieved by the proposed method is limited. Given that the method incorporates temporal information, velocity estimation, and depth estimation tasks, its performance enhancement compared to the CRN framework is relatively modest.

**Limitations:**

Please consult the question section for further information.

---

> ### Author Rebuttal · Authors · 2024-08-06
>
> **W1: The paper lacks a detailed description of the loss function used in the proposed framework.**
>
> The total loss function used in CRF-Fusion consists of five loss terms: a standard 3D object detection loss term and four additional loss terms derived from different head networks within our model. The total loss $ L_{total}$ is given by
>
> $ L_{total} = L_{det} + \lambda_{depth} L_{depth} + \lambda_{seg} L_{seg} + \lambda_{vel} L_{vel} + \lambda_{occ} L_{occ}, $
>
> where $L_{det}$ is the 3D object detection loss, $L_{depth}$ is the loss from the Depth Prediction Head in MVF, $L_{seg}$ is the loss from the Perspective-View Semantic Segmentation Head in MVF, $L_{vel}$ is the loss from the Velocity Prediction Head in MFE, and $L_{occ}$ is the loss from the Object Occupancy Prediction Head in MFE. The parameters $\lambda_{depth}$, $\lambda_{seg}$, $\lambda_{vel}$, and $\lambda_{occ}$ are the  weights for the corresponding loss terms.
>
> The specific descriptions for each loss component are as follows. The Depth Prediction Loss uses binary cross-entropy loss for depth estimation with a weight of $\lambda_{depth} = 3.0$ following the approach used in BEVDepth. For the Perspective View Segmentation Loss, we also employ the binary cross-entropy loss with a weight of $\lambda_{seg} = 25$. The Velocity Prediction Loss, which handles velocity $(v_x, v_y)$ and orientation prediction, utilizes Mean Squared Error (MSE) with a weight of $\lambda_{vel} = 1$. Finally, the BEV Object Occupancy Loss uses Binary Focal Loss for foreground and background segmentation with a weight of $\lambda_{occ} = 30$.
>
> We will add the detailed information on the loss function in the final version.
>
>
> **W2: No experimental evidence is provided that the MVF module enhances depth prediction.**
>
> We conducted additional experiments for analyzing the impact of the RCA component in MVF module on depth prediction accuracy.  Our experimental results show that utilizing RCA significantly improves depth prediction accuracy in Mean Square Error (MSE). The table below provides the detailed results.
>
> | Methods       | Depth MSE (m) |
> |---------------|:-------------:|
> | Ours w/o RCA  |      4.3      |
> | Ours w RCA    |      3.8      |
>
>
> **W3: The paper does not clarify how the velocity estimation in the MFE module is supervised during the learning process.**
>
> The velocity values predicted by the MFE module are supervised with the GT velocity values through MSE loss. Details on the loss function used in CRT-Fusion are provided in our response to the comment W1. The MSE loss measures errors in both the magnitude of velocity in the x and y axes, as well as the orientation value of the velocity.
>
> **W4: The overall improvement by the proposed method is limited compared to the CRN framework. Given that the method incorporates temporal information, velocity estimation, and depth estimation tasks, its performance enhancement compared to the CRN framework is relatively modest.**
>
> Table 2 in our manuscript shows that our CRT-Fusion achieves a performance gain of 3.2\% in NDS and 1.4\% in mAP over CRN. These gains can be considered substantial in the highly competitive nuScenes BEV object detection benchmarks! Please note that like our CRT-Fusion, the CRN method also used both depth estimation and utilized temporal information. Therefore, the performance gains achieved by our proposed method are attributed to both radar-assisted depth prediction and velocity-driven feature alignment. This highlights the significant impact of our main ideas.

---

### Official Review · Reviewer_6YHD · 2024-07-10

**Soundness:** 3
**Presentation:** 3
**Contribution:** 2
**Rating:** 5
**Confidence:** 4

**Summary:**

The paper proposes CRT-Fusion that fuses radar and camera inputs for 3D object detection in BEV space. A radar-camera azimuth attention module is proposed for improving image features for better motion awareness, which further improves the quality of image BEV features. A motion-guided temporal fusion is introduced to align dynamic objects in history BEV feature maps with their positions in the current BEV feature map, using predicted object presence information and predicted velocities. The authors conduct experiments on the nuScenes dataset and prove the effectiveness of the proposed model.

**Strengths:**

1. The proposed framework is concise and straightforward, while showing good performance on nuScenes benchmark
2. The proposed RCA first decouples the image feature map to row and column features, and only applies cross-attention on column features, which enhances image features with a relatively low computation cost.
3. The motion guided temporal fusion mitigates the BEV feature misalignment issue of moving objects across different timestamps, and improves the 3D object detection performance.

**Weaknesses:**

1. The proposed temporal fusion requires aggregating multiple feature maps of different timestamps in a recurrent way. The inference latency and required GPU memory will grow linearly with the number of past frames.
2. The multi-view fusion section needs more details. It's unclear how the radar BEV feature map is obtained and how the gated fusion is performed. It's better to include some equations for illustration.
3. The equation 7 for motion-guided temporal fusion seems to be incorrect. Further details for generating the shited feature map are needed. For example, let's consider this situation: the velocity predictions at (x1, y1) and (x2, y2) are (Δx1, Δy1) and (Δx2, Δy2) respectively, and x1+Δx1 = x2+Δx2, y1+Δy1=y2+Δy2, what will the shifted feature map be at (x1+Δx1, y1+Δy1)?

**Questions:**

In Table 1, I notice that CRT-Fusion-light has lower mATE, mASE, mAOE, mAVE, mAAE compared to CRT-Fusion, which is unexpected. It would be beneficial if the authors could provide a detailed explanation or revisit the experimental setup to clarify these results.

---

> ### Author Rebuttal · Authors · 2024-08-06
>
> **W1: The inference latency and required GPU memory will grow linearly with the number of past frames.**
>
> Our model utilizes a memory bank structure where previous BEV features obtained through a cascade of backbone, MVF, and MFE are stored in the buffer to reduce redundant computations. Thus, only the computation of MGTF increases  as the number of frames increases. This design significantly reduces the computational overhead for temporal fusion.
>
> As illustrated in Figure 2 of our PDF file, our model uses only 3.744 GB of GPU memory, and achieves a latency of 67.1ms, which demonstrates superior performance to the existing methods.
>
> **W2: The multi-view fusion section needs more details. It's unclear how the radar BEV feature map is obtained and how the gated fusion is performed.**
>
> We apologize for the lack of details in the MVF (multi-view fusion) section.
>
> First, the radar BEV feature was obtained by applying a PointPillars-based backbone network.
>
> MVF combines feature maps obtained from each modality using the Gated Fusion Network. The Gated Fusion Network computes the combining weights that adaptively adjust the contributions of the two modalities.
> Let $F_r$ be the radar BEV feature and $F_c^{BEV}$ be the camera BEV feature map. The equations for obtaining the combining weights are given by
>
> $W_r = \sigma(\text{Conv}_r(F_r \oplus F_c^{BEV}))$
>
> $W_c =  \sigma(\text{Conv}_c(F_r \oplus F_c^{BEV})),$
>
> where  $\sigma$ denotes the sigmoid function, and $\text{Conv}_r$ and $\text{Conv}_c$ are convolutional layers.
> The final fused BEV feature map is obtained by
>
> $F_{fused} = (W_r \times F_r) \oplus (W_c \times F_c^{BEV}),$
>
> where  $\times$ denotes element-wise multiplication operation and $\oplus$ indicates element-wise summation operation.
> We will add this explanation in the final version.
>
> **W3: The equation 7 for motion-guided temporal fusion seems to be incorrect. Further details for generating the shifted feature map are needed.**
>
> As the reviewer mentioned, there may be cases where feature maps overlap when shifted. To address this, we apply averaging operation on the overlapping features. We will comment on this point in the final version of our paper.
>
> We can revise the equation 7 as folows:
> For each coordinate $(x, y)$ in the BEV feature map $B_{t-k}$, the feature value $B_{t-k}(x, y)$ is shifted based on the corresponding velocity vector $M_{t-k}(x, y) = [\Delta x, \Delta y]$ if the magnitude of the velocity exceeds a specified threshold $\tau_v$. The shifted feature maps are obtained as
>
> $B'_{t-k}(x, y) = \frac{1}{|S(x,y)|}\sum_{(i,j) \in S(x,y)} {B}_{t-k}(i, j)$
>
> where
>
> $S(x,y) = \{(i,j): x= i+\lfloor \Delta x \rceil, y= j+ \lfloor \Delta y \rceil , |M_{t-k}(i, j)|> \tau_v \}$
>
> where  $|S(x,y)|$ denotes the cardinality of $S(x,y)$ and $\lfloor \cdot \rciel$ denotes the rounding operation.
>
> **Q1: An explanation is needed for CRT-Fusion-light's performance compared to CRT-Fusion in Table 1.**
>
> It is not fair to compare CRT-Fusion with CRT-Fusion-light&#8224; since CRT-Fusion did not employ CBGS augmentation.
> Please compare CRT-Fusion&#8224; with CRT-Fusion-light&#8224; since CRT-Fusion&#8224; employed CBGS augmentation.

---

> > ### Comment · Reviewer_6YHD · 2024-08-09
> >
> > The authors successfully address my questions. I would like to keep my original rating.

---

> ### Author Response · Authors · 2024-08-08
>
> During the editing process, our session closed, causing some parts of the equations in W3's response to be omitted. We apologize for this. We are now adding the explanation for the relevant parts:
>
> The shifted feature maps are obtained as
>
> $B' _ {t-k}(x, y) = \frac{1}{|S(x,y)|}\sum_{(i,j) \in S(x,y)} {B}_{t-k}(i, j)$
>
> where
>
> $S(x,y) = \\{(i,j): x= i+\lfloor \Delta x \rceil, y= j+ \lfloor \Delta y \rceil , |M_{t-k}(i, j)|> \tau_v \\}$
>
> where  $|S(x,y)|$ denotes the cardinality of $S(x,y)$ and $\lfloor \cdot \rceil$ denotes the rounding operation.

---

### Official Review · Reviewer_NhQh · 2024-07-11

**Soundness:** 2
**Presentation:** 3
**Contribution:** 2
**Rating:** 4
**Confidence:** 3

**Summary:**

The paper talks about multi sensor fusion using Motion Information for Bird’s Eye View Object
Detection. Authors have presented a multiple step approach to improve the object detection on nuScenes dataset. The proposed approach comprises of three parts; Multi-View Fusion (MVF) that enhances depth prediction by leveraging radar features to improve image features before fusing them into a single BEV vector; a Motion Feature Estimator (MFE) step that estimates pixel-wise velocity information and Motion Guided Temporal Fusion (MGTF) step that iteratively aligns and fuses feature maps across multiple timestamps. The proposed approach is tested on the latest nuScenes dataset for object detection and obtains state-of-the-art results on the dataset surpassing many publicly listed approaches.

**Strengths:**

1. The paper is easy to read and follow. Authors have described each step carefully and mentioned proper mathematical formulation for each step.
2. The motivation and improvements in the performance is clear. In many cases the proposed framework archives state of the art results.
3. Ablation study is also very well supported. Authors have included results for baseline modules and how each of their component improves the performance of object detection.

**Weaknesses:**

1. I believe that the technical novelty of the paper is somewhat limited. The idea of MVF fusion, i.e. Radar and Camera image feature fusion has been studied well in the past.
2. The paper lacks qualitative results completely. No where in the main paper have the authors presented any qualitative analysis. This makes it hard to interpret where the proposed approach performs better than current methods and more importantly where the proposed approach fails.
3. Overall approach with multiple components, although produces state-of-the art results, but is nearly impractical to scale or even productionize in a real world setting. Authors haven’t mentioned anywhere in the paper about this limitation (only described it briefly in the Supplementary section). The limitations of the current approach should be more broadly presented in the main paper, i.e. if the number of frames increase, the model will have high computational cost and hence not scalable.

**Questions:**

1. Can authors provide a small description of how this model can be made a bit efficient such that this can be deployed in the real world?
2. Figure 2 is not very clear. For the MVF (bottom left side) figure, what does the middle gray box represent?
3. Line 135 states an LSS approach, but that hasn’t been mentioned anywhere. What is LSS?
4. Line 137 states “existing state-of-the-art…”, can authors please cite which work they are referring to here?
5. The calculation of GT for MFE is not very clear. Why do we use the input for MVF to compute the GT for MFE (line 199)?
6. For line 201, how did the authors determine the value for T_{iou}? Why did they consider 0.5? Was there some ablation study conducted?
7. For Equation 7, it’s not very clear what B^_{t-k} is? Also how was the value for T_{v} chosen here?
8. Table 1 here refers to TTA. What is TTA?
9. Can the authors provide some qualitative results? And compare the existing work with the proposed approach to show what are the strengths of their approach? Just listing down metrics may not be motivating enough to see where the proposed approach does best.
10. Line 283 states that CRT does not do well in Sunny conditions which is a bit strange since sunny weather conditions should be ideal for object detection. Can the authors provide some explanation here?
11. For table 6, RCA has worse performance as compared to RGVT on mATE. Can authors address why is that the case?

**Limitations:**

Authors have presented the limitations of the work (which is a good thing), but this has been mostly discussed in the supplementary section (which is outside of the main paper). I would encourage the authors to include a brief summary of the limitations in the main paper as well.

---

> ### Author Rebuttal · Authors · 2024-08-06
>
> **W1: The novelty of the paper is limited.**
>
> We are sorry that the reviewer does not appreciate the contribution of our study, to which we have dedicated much effort. We believe that our work can make a substantial contribution to enhancing the effectiveness of temporal fusion in the field of radar-camera based 3D object detection. While several radar-camera fusion methods have been previously studied, our research primarily focuses on an effective temporal fusion approach for camera-radar integration. Unlike existing methods that merely concatenate features from previous frames, our method achieves spatial feature alignment using predicted velocity information. This novel approach has not been proposed in previous methods such as CRN and BEVFusion-R. Our method performs feature-level temporal alignment and fusion, resulting in significant performance gains over existing techniques.
>
> **W2: The paper lacks qualitative results.**
>
> We have presented qualitative results in Figure 1 of the attached PDF file. In the final version, we will add this comprehensive qualitative analysis.
>
> **W3: The approach is impractical for real-world deployment and lacks sufficient limitations analysis.**
>
> Although our model achieves state-of-the-art performance, its latency and memory usage are comparable to other methods, making it suitable for practical implementation. We have also presented a lightweight version, CRT-Fusion-Light, which meets real-time requirements for deployment in real-world settings.
>
> We have discussed the limitations of our model in the Author Rebuttal section  Q3.
>
> **Q1: Can authors provide a small description of how this model can be made more efficient for real-world deployment?**
>
> The computation time of our model can be optimized using model compression techniques or reducing the model capacities. CRT-Fusion-light, presented in Table 1 of the manuscript, is an optimized version of CRT-Fusion that yields improved FPS with minimal performance drop. Please note that CRT-Fusion-light achieves 20.5 FPS, potentially satisfying real-time requirements.
>
> **Q2: Figure 2 is not very clear. For the MVF (bottom left side) figure, what does the middle gray box represent?**
>
> In Figure 2, the middle gray box in the MVF represents the intermediate feature, $\hat{F}_c$, as shown in Figure 3
>
> **Q3: Line 135 states an LSS approach, but that hasn’t been mentioned anywhere.**
>
> We apologize for not providing the introduction to LSS. The LSS (Lift, Splat, Shoot) method is a widely used operation for transforming multi-view 2D images into BEV features in camera-based 3D perception. This method was introduced in the ECCV 2020 paper `Lift, Splat, Shoot: Encoding Images From Arbitrary Camera Rigs by Implicitly Unprojecting to 3D'. LSS predicts depth values for each pixel in 2D images and converts multi-view camera features into BEV feature  map through view transformation. We will add detailed information about LSS in the final version.
>
> **Q4: Line 137 states “existing state-of-the-art…”, can authors please cite which work they are referring to here?**
>
> The 'existing state-of-the-art' mentioned in Line 137 refers to the CRN method. We will mention CRN in the final version.
>
> **Q5: The calculation of GT for MFE is not very clear. Why do we use the input for MVF to compute the GT for MFE (line 199)?**
>
> We apologize for any confusion regarding the calculation of GT for MFE. We did not use the BEV feature map $B_{t-k}(x,y)$ in generating GT velocity.  In equation 4, we calculated the IoU ratio $r(x,y)$ between the physical box whose size corresponds to the pixel at $(x,y)$ and the 3D object boxes projected onto the BEV domain. If this IoU ratio was higher than $\tau_{iou}$, we considered this pixel as positive and assigned the GT velocity for supervision.
>
> **Q6: For line 201, how did the authors determine the value for $\tau_{iou}$ Value? Why did they consider 0.5? Was there some ablation study conducted?**
>
> The threshold $\tau_{iou}$ was determined empirically. Below are the performance of CRT-Fusion obtained with several values of $\tau_{iou}$.
>
> | $\tau_{iou}$ | NDS  | mAP  |
> |------------|------|------|
> | 0.3       | 56.9 | 49.3 |
> | 0.5   | **57.2** | **50.0** |
> | 0.7       | 57.0 | 49.7 |
>
> **Q7: For Equation 7, it’s not very clear what $B_{t-k}$ is? Also how was the value for $\tau_{v}$ chosen here?**
>
> The notation $B_{t-k}$ is the BEV feature map obtained from the MVF module.
>
> The threshold $\tau_{v}$ was determined experimentally. Our ablation study for $\tau_{v}$ was provided in Table 9 (c) of the Supplementary Materials.
>
> **Q8: What is TTA?**
>
> TTA stands for Test Time Augmentation, a method that applies various augmentations to test data to improve model prediction.
>
> **Q9: Can the authors provide some qualitative results? And compare the existing work with the proposed approach to show what are the strengths of their approach?**
>
> We have added qualitative results of our CRT-Fusion in Q2 of the Author Rebuttal. Our detailed comparison with CRN was provided in Figure 1 of the PDF file attached.
>
> **Q10: Line 283 states that CRT does not do well in Sunny conditions which is a bit strange since sunny weather conditions should be ideal for object detection.**
>
> In sunny conditions, our model achieves 54.7\% mAP, nearly identical to CRN with only a 0.1\% difference. In contrast, we see notable improvements in more challenging conditions like rainy weather and nighttime. The slight variation in sunny conditions is within the margin of normal experimental variance and does not indicate a significant trade-off. Instead, it demonstrates that our model maintains strong performance in good conditions while offering substantial improvements in challenging scenarios.
>
> **Q11: For table 6, RCA has worse performance as compared to RGVT on mATE.**
>
> Our RCA module result in only 0.002 drop in mATE compared to RGVT. Unfortunately, we were unable to figure out why RCA did not improve mATE.

---

### Author Rebuttal · Authors · 2024-08-06

Thank you for your thoughtful reviews. We have provided responses to the common questions raised by the reviewers below. Additionally, we have attached PDF files presenting qualitative results, as well as the analysis of GPU memory usage, latency, and performance relative to the number of frames.

**Q1: A comparative analysis with existing research is needed as the number of frames increases.**

Our model utilizes a memory bank structure where previous BEV features obtained through a cascade of backbone, MVF, and MFE are stored in the buffer to reduce redundant computations. Thus, only the computation of MGTF module increases  as the number of frames increases. This design significantly reduces the computational overhead for temporal fusion.

Table 1 and Figure 2 in the attached PDF file present the analysis of performance, latency, and GPU memory usage of CRT-Fusion as the number of past frames increases. (The hardware configuration included an NVIDIA RTX 3090 GPU and an Intel Xeon Silver 4210R CPU processor.) The trend of CRT-Fusion is compared with that of CRN.  For a fair comparison, we used the same camera backbone, radar backbone, and input image size. Our model achieved better performance in terms of mAP and NDS across all frames. Figure 2 also demonstrates that our model's GPU memory usage is consistently lower, and the latency of CRT-Fusion increases more slowly with the number of past frames compared to CRN. We will add these results in the final version.

**Q2: The paper lacks qualitative results.**


We conducted a qualitative analysis of CRT-Fusion compared to CRN. Figure 1 in the attached PDF file shows the qualitative results in different scenarios. In Figure 1(a), CRT-Fusion accurately detects objects that CRN misses. In Figure 1(b), CRT-Fusion better predicts vehicle orientations and centers. Figure 1(c) also highlights CRT-Fusion's improved accuracy over CRN. We will include this qualitative analysis in the Supplemental Materials of the final version.


**Q3: The analysis of limitations is insufficient.**

Here is our discussion about the limitations of our study, which will be added in the final version:
While our model has achieved significant performance gains over the baseline, the computation time increases with the number of previous frames used for temporal fusion. This limitation means our method cannot accommodate as many previous frames as desired due to hardware constraints. This issue might stem from the parallel fusion structure used to combine BEV features. One potential remedy is to adopt a recurrent fusion structure that combines BEV features temporally in a recurrent fashion. By doing so, the computational feasibility can be maintained while incorporating BEV features from a long-term historical frames. In the future, we will explore the recurrent fusion architecture for CRT-Fusion.

**Q4: What is $B_{t-k}(x, y)$ in equation 4?**

We apologize for any confusion regarding the calculation of GT for MFE. It is confusing to use $B_{t-k}(x,y)$ in Equation 4.

In Equation 4, $r(x,y)$ represents the Intersection over Union (IoU) ratio between the physical box corresponding to the pixel at a position $(x,y)$ and the 3D object boxes projected onto the BEV domain. Pixels with an IoU ratio exceeding $\tau_{iou}$ are classified as positive and are assigned the GT velocity and GT occupancy state for supervision. We revised the equation 4 by replacing $B_{t-k}(x,y)$ with $H(x,y)$:

Define the IoU ratio $r(x, y)$ for pixel $(x, y)$ as

$r(x, y) = \frac{|H(x, y) \cap \mathcal{P}(\mathcal{G})|}{|H(x, y) \cup \mathcal{P}(\mathcal{G}) |}$

where \( H(x, y) \) is the physical bounding box whose size corresponds to the pixel at $(x, y)$, $\mathcal{G}$ is the set of 3D object boxes, and $\mathcal{P}(\mathcal{G})$ is the projection of these boxes onto the BEV domain.

---

### Comment · Reviewer_NhQh · 2024-08-12
**Response to rebuttal**

Thank you authors for presenting the rebuttal documents. I have read through the comments and the attached PDF. Since most of my concerns are addressed, I am okay in changing my rating from "borderline reject" to "borderline accept". There are still some concerns on the complexity of the solution but given that most of the issues have been addressed, I am inclined towards borderline acceptance here.

---

### Decision · Program_Chairs · 2024-09-25

**Decision:**

Accept (poster)

**Comment:**

Details TBD, needs discussion.

Scores are very borderline, 4, 5, 4, 5, 6.